# Design and Implementation of Quality Information Management System for Modular Construction Factory

**Jiwong Shin and Byungjoo Choi \***

Department of Architectural Engineering, Ajou University, Suwon 16499, Korea; sgo802@ajou.ac.kr
* Correspondence: bchoi@ajou.ac.kr

**Abstract:** Modular construction has been gaining increasing attention from industry and academia as a solution to the limitations of the traditional on-site orient production systems in the construction industry. Various attempts have been made to improve modular construction performance. However, while previous studies have attempted to enhance the productivity of modular construction, attempts to improve the efficiency of quality management in modular construction have been limited. Moreover, the quality management practices in a modular factory still rely on document-oriented quality information management, which is inefficient. Therefore, this study aims to develop a quality information management system to improve quality information management during module manufacturing. Accordingly, quality information during module manufacturing has been standardized using integration definition for process modeling, and system functions are defined using standardized quality information. The developed modular factory quality information management system includes module information and production-type management, material management, and module production management. The practicability and validity of the developed system were examined by accredited tests and certification laboratory and modular construction experts. The developed system is expected to contribute to improving the existing inefficient quality management process of module manufacturers by providing an integrated and systematic method to manage quality information generated during manufacturing.

**Keywords:** modular construction factory; quality management; quality information management system; off-site construction

## 1. Introduction

Quality management refers to all management activities aimed at achieving the intended use of a product and satisfying user requirements. It seeks to maintain and improve quality by preventing defects. This is achieved by identifying and managing the causes of defects in manufacturing. Construction quality management can be defined as the management of construction according to the design documents and contract with the owner [1]. Construction quality management failures affect various parts of a construction project. Further, construction work that does not adhere to the proposed design often leads to reconstruction, which negatively affects the cost and construction period [2]. In addition, insufficient quality management in construction works can lead to poor construction and, consequently, large-scale casualties [3].

With construction projects becoming larger, more complex, and more advanced, the importance of quality information for effective project management has increased. The accumulation of quality information can lower the defect rate in construction projects by increasing the efficiency of quality management. A reduction in the defect rate positively affects cost and schedule in construction projects by decreasing the rework rate [4]. Further, with construction projects growing in size and complexity, the amount of quality information to be managed has also increased. Quality information is generated in all phases of construction projects, such as during contract change, design change, and nonconforming

construction management, in addition to physical inspections for building components, such as construction and material inspections [5]. The construction industry introduced (and continues to utilize) information technology to effectively manage large amounts of quality information [6,7]. For example, technologies such as continuous acquisition and life-cycle support (CALS) and project information management system (PIMS) have been widely adopted in the construction industry.

Meanwhile, modular construction has emerged to overcome the limitations of the existing site-oriented construction method [8]. As a type of off-site construction (OSC), it involves modularizing a building into a panel- or volumetric-type unit (Lawson et al., 2014). It combines the existing construction method of site-oriented operation with the production methods of the manufacturing industry [9,10]. Specifically, modular construction involves transporting and installing building modules manufactured in a factory to the site. The current site-oriented construction method exhibits low productivity; it is difficult to utilize production automation equipment owing to the narrow workspace [11,12]. Further, because most construction activities are performed outdoors, site-oriented construction methods are greatly affected by weather conditions [13]. In contrast, it is easy to use automated facilities in a modular construction factory, and the influence of weather conditions is not significant because the main processes are performed indoor factories [14,15]. Furthermore, modular construction can reduce construction time by simultaneously executing module manufacture in the factory and on-site construction processes, such as excavation and foundation construction [9,16]. Therefore, modular construction can improve productivity by approximately 60% compared to the site-oriented construction method [17]. In addition, modular construction is receiving a lot of attention as the future of construction technology because it is better than site-oriented construction work in terms of safety, waste reduction, and quality [17,18]. Modular construction can be considered as a process between manufacturing and construction. Unlike general manufacturing, which focuses on mass production of small items, modular construction is a project-based production system. However, modular construction's factory-based production system is also different from that of general construction. Owing to such unique characteristics of module manufacturing, various attempts have been made to find better methods of improving the performance of module manufacturing. Goh and Goh [19] proposed a method to improve the productivity of modular construction by applying lean production. Accordingly, they suggested implementing total quality management (TQM), utilizing the E-Kanban Just-in-Time system, multi-skilled labor, and using robots. A comparative analysis of the productivity of the existing and new methods through simulation confirmed that the cycle time was reduced by up to 81.27% and the work in progress by up to 74.30% using the proposed method. Lee and Lee [20] developed a BIM-based digital twin framework and proposed a method for optimizing the transportation of modules from the factory to the site. The developed framework improved the productivity of the modular project by automatically identifying potential risk factors in the transportation process and providing an optimal transport route.

Although studies on modular construction are continuously being conducted, research on its quality management aspect is insufficient. Most of the research to date has focused on improving the productivity of modular construction. Goh and Goh [19] mentioned the need for TQM during the manufacturing process in module factories. However, they could not suggest a specific quality management plan for improving productivity. Although the importance of quality information in the construction industry is increasing, utilizing quality information in modular construction has not been researched. The main difference between modular construction and site-oriented construction is the factory manufacturing method employed in the former. Characteristically, the manufacturing industry repeatedly produces products according to a set production line [21]. Further, it is easy to manage and utilize quality information because related information is also generated sequentially according to a set manufacturing process [22,23]. In contrast, it is difficult to collect and utilize quality information in site-oriented construction work due to complexity and uncertainty, such as sudden actions of workers and changes in the weather [24]. Therefore, quality

information can be better utilized in modular construction than in site-oriented construction. Despite the potential of using quality information in modular construction, studies on the management and utilization of quality information have not yet been conducted. In this study, interviews with experts working at module manufacturing factories were conducted to elucidate the quality information management status at the manufacturing stage in a module factory. The interviews were conducted thrice, targeting two different modular construction companies. The two companies are leading modular construction companies in South Korea. Two interviewees with more than ten years of experience in modular construction participated in the interviews from each company. The first one was a paper-based interview focusing on the basic procedures of quality management in module manufacturing factories. Based on the responses from the first interview, a second in-person interview was conducted to clarify the current problems in quality management processes in module manufacturing factories. The last interview was conducted to confirm the issues in the quality management processes of module manufacturing factories identified herein. Both manufacturers managed the quality information generated during the module manufacturing process by relying on documents. Document-oriented quality information management is associated with various problems, such as the loss of written documents and duplicate documents, resulting in inefficient quality management. In addition, large amounts of quality documents were produced during one modular construction project. Because many projects were simultaneously conducted, the amount of documents generated within the module manufacturing company was high. However, there was no method of systematically storing and managing quality information during module manufacturing. Notably, quality information cannot be utilized properly if a separate quality information management plan is not prepared.

This study aims to develop a web-based quality information management system to improve quality information management in the module manufacturing process. Using it, module manufacturers can break away from the existing document-centered quality information management. It facilitates effective management and utilization of the quality information generated during module manufacturing. Regarding system development, this study analyzed module manufacturing in detail. Afterwards, quality information generated in each subdivided task was classified according to its characteristics. Because there is no research on the utilization of quality information in module manufacturing, this study will elucidate the quality information generated in the module manufacturing process. Furthermore, the system can positively affect the module quality level as well as improve the quality control work efficiency.

## 2. Methods

The quality information management system was developed in the following order: (1) collection of quality data generated during manufacturing in a modular factory, (2) standardization of quality information, (3) defining the system function, (4) system development, and (5) system verification. Figure 1 shows a schematic of the system development process.

1.  Data collection was performed to identify key quality information in the module manufacturing process, and key quality information generated at each stage of the process was derived.
2.  To integrate the various types of quality data generated in module manufacturing into a single management system, information standardization was performed using integration definition for process modeling (IDEF0).
3.  The overall system design direction was determined based on the previous content, and the main functions of the system were defined accordingly.
4.  A quality information management system was developed according to the defined functional contents.
5.  The usefulness of the developed system was validated by accredited certification laboratory test and modular construction experts.

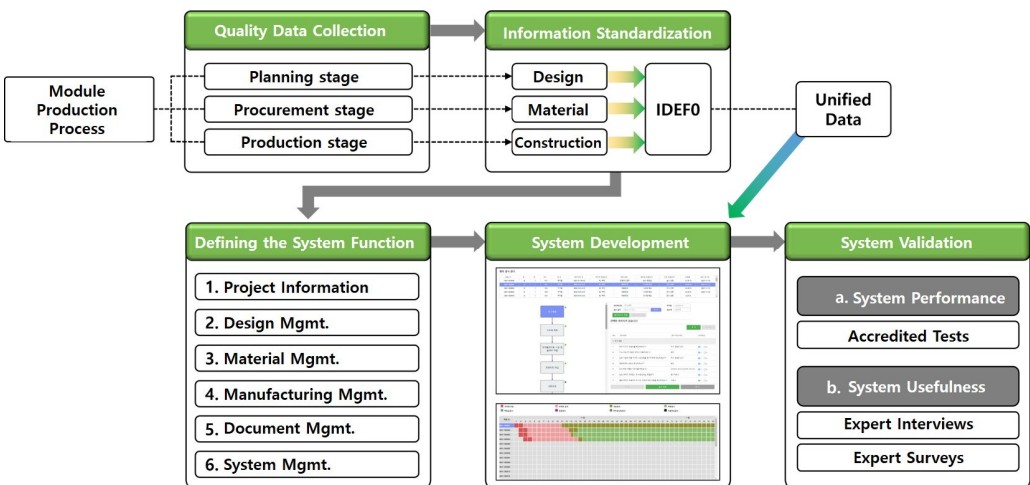

**Figure 1.** Framework of the development of quality information management system for modular factory.

## 3. Design of Quality Information Management System

### 3.1. Module Production Process Analysis

An abundant amount of quality information is generated during module manufacturing. It is generated from all activities that can affect modular quality, such as physical inspection and testing of building components, as well as approval management of material suppliers, management of manufacturing personnel, and management of design documents. Module manufacturing in the factory is conducted in connection with various tasks, such as design, production, and procurement. Quality information generated in the module production stage is also directly or indirectly influenced by quality information generated in other tasks. For example, information related to the main materials and production of the module may vary depending on the quality information generated in the design phase. Thus, the relationship among different types of quality information is complicated because quality information is generated in all tasks during module manufacturing, and the generated quality information affects other information. Therefore, to effectively manage quality information, it is necessary to identify and classify the types and characteristics of tasks that generate quality information during module manufacturing. To elucidate the role of quality management during module manufacturing, we conducted an analysis of modular manufacturers' documents related to quality management, a literature review, and expert interviews. Figure 2 shows a schematic of the quality management process during module production.

The quality management process in module production can be divided into a prepreparation stage and a module assembly stage based on the time point in module assembly. The former can be further divided into planning and procurement stages. The planning stage starts with the distribution of the completed design documents. A basic management plan for module production is established based on the contents in these documents. The module production management plan refers to the overall management plan for quality management, production management, material import, etc. The initial plan becomes more detailed, according to the actual production environment, as production progresses. After the planning stage, the procurement stage proceeds according to the basic management plan. This phase primarily involves bringing in the main materials selected in the planning stage. The main tasks include selection of material suppliers, making material purchase requests, and material ordering. The factory production stage primarily involves inspection of incoming materials and construction works. After the inspection of the final activity, the module production quality management process is completed by shipping the finished module.

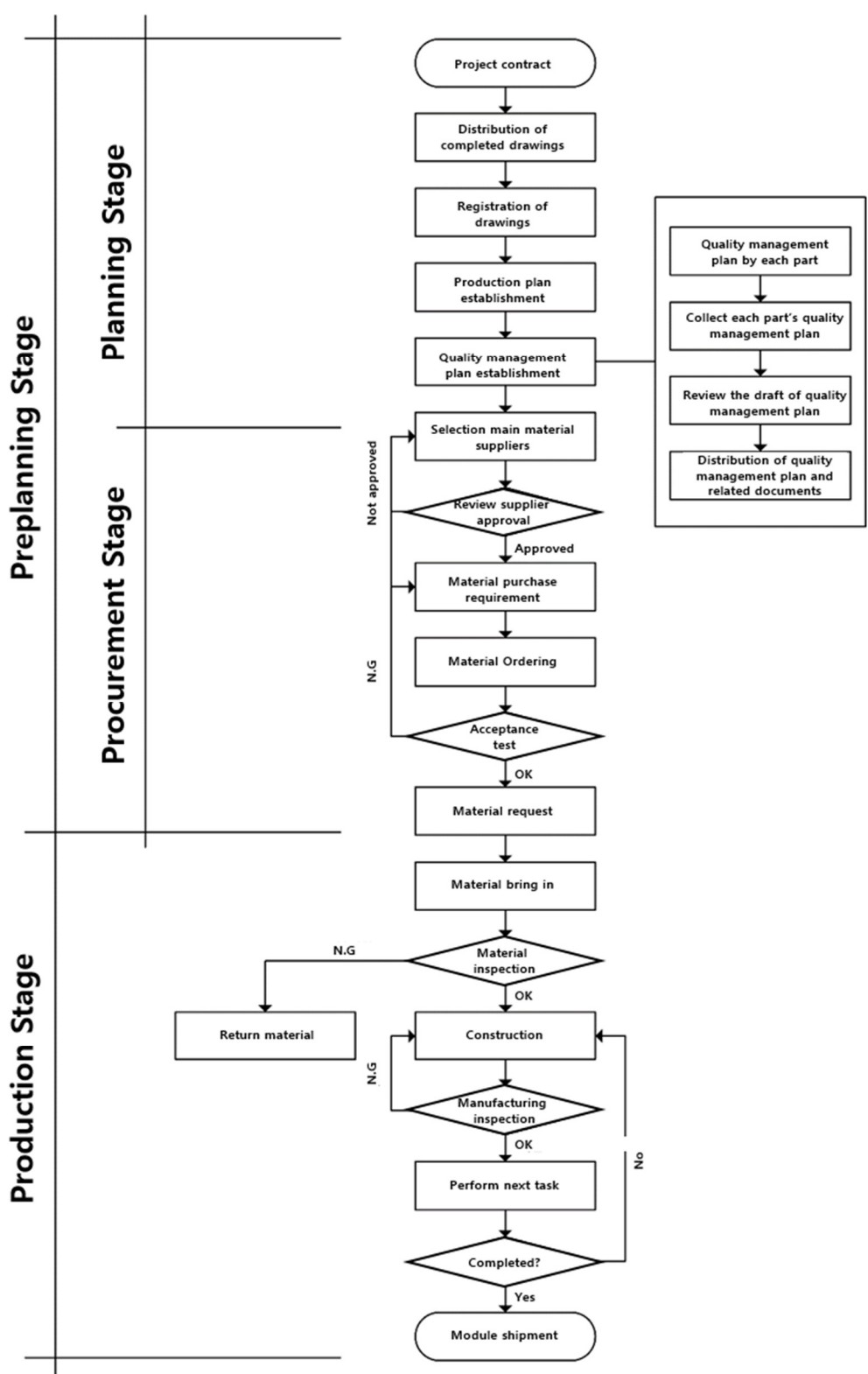

**Figure 2.** Quality management process in module production.

### 3.2. Standardization of Quality Information

As previously mentioned, quality information is generated at each stage of module manufacturing. Quality information has different characteristics according to the objective of quality management activities at each stage. Information standardization is required to manage information with different attributes in an integrated way through a single

platform [25,26]. It refers to standardizing different types of quality information generated in each process of module production into a specific data format. Quality information in the module manufacturing process is sequentially generated according to the module manufacturing process. IDEF0, one of the process modeling techniques, can comprehensively represent the input and output information, control (i.e., a set of conditions required for a function), and mechanism (i.e., means or resource required to transform input to output) of each function according to the workflow. It is considered an easy and effective method of expressing complicated processes and functions [27]. Therefore, many studies have used IDEF0 for information system development [28]. Gingele et al. [29] applied IDEF0 to the quality management practices of manufacturers and demonstrated that IDEF0 is also effective in dealing with the international standard for quality management systems (ISO 9001). In this study, the quality information of the module manufacturing process was standardized through the IDEF0 method. For effective information standardization, the division between each stage of the module manufacturing process must be distinct. If the stage division of the production process is unclear, determining which information is input/output at which stage in detail and how it is influenced by other information becomes difficult. This study analyzed the object of quality management activities at each stage of the module production process to clarify the information standardization results. Accordingly, it was confirmed that the manufacturing plan establishment process was centered on design, a procurement stage focused on materials, and a factory manufacturing stage focused on construction. Therefore, information standardization was conducted by dividing the factory manufacturing stage into tasks related to design, materials, and construction [30,31]. Table 1 shows the quality information for each production task identified through information standardization.

### 3.2.1. Standardization of Design Information

Figure 3 shows the standardization result of design quality information during module manufacturing. Design-related quality control tasks can be expressed in the following steps: (A01) preparation of design documents, (A02) review of design (draft), and (A03) preparation of factory production drawings. The design document preparation stage requires basic project information, such as the owner requirements and the project outline. Specifically, in the design document preparation stage, information on the area and height of the modular building, the type of interior/exterior material of the building, etc., is input. Further, basic design document details, such as design drawings, specifications, structural calculations, and bill of quantities, are generated by the design team led by the architect. The basic form of each manufactured module is determined according to the information generated in the design document preparation stage. When completed, the relevant departments, such as the engineering and manufacturing teams at the manufacturer, review the prepared design documents. This review stage involves examination of discrepancies between the contents of the design documents and interference in construction to determine the probability of presence of defects based on the design documents. During design review, new design document information is generated depending on the number of revisions. Information about the production drawing for manufacturing the module is generated from the reviewed design documents by the manufacturer's engineering and manufacturing teams. Factory-manufacturing drawings are created at a more detailed level than the previously created design drawings in that they are created to ensure constructability and precision in the manufactured module. The more accurate the factory manufacturing drawings are, the less the defects in the module manufacturing process and risk of reconstruction.

**Table 1.** Quality information generated during each production task.

| Division | Input | Output | Control | Mechanism |
|---|---|---|---|---|
| A.<br>Design | 1. Contract agreements<br>2. Owner's requirements<br>3. Basic project information | 1. Design drawing<br>2. Specifications<br>3. Structural calculations<br>4. Bill of quantities<br>5. Factory production drawings | 1. Contract document<br>2. Relevant regulations<br>3. Design work procedure<br>4. Document management procedure | 1. Plant manager<br>2. Design team<br>3. Owner<br>4. Production team<br>5. Engineering Team |
| B.<br>Material | 1. Owner's requirements<br>2. Basic project information<br>3. Schedule<br>4. Required quantity<br>5. Supplier approval request<br>6. Material performance test results<br>7. Application for bringing in material | 1. Material supply plan<br>2. Review result of supplier approval request<br>3. Purchase order<br>4. Acceptance inspection result document<br>5. Material approval documents<br>6. Invoice for incoming materials<br>7. Material inspection result document | 1. Contract document<br>2. Design drawings<br>3. Specification<br>4. Project Schedule<br>5. Bill of quantities<br>6. Relevant regulations<br>7. Document management work procedure<br>8. Inspection and test plan | 1. Owner<br>2. Plant Manager<br>3. Production team<br>4. Material Supplier |
| C. Construction | 1. Contract Agreement<br>2. Owner's requirements<br>3. Basic project information<br>4. Material supply plan<br>5. Production manpower status<br>6. Status of major facilities<br>7. Request for inspection and test | 1. Construction and production plan<br>2. Project schedule<br>3. Inspection and test plan<br>4. Inspection Checklist<br>5. Inspection result document<br>6. Test result document | 1. Contract document<br>2. Design drawings<br>3. Specification<br>4. Bill of quantities<br>5. Material supply plan<br>6. Document management work procedure<br>7. Relevant regulations | 1. Owner<br>2. Plant Manager<br>3. Production team |

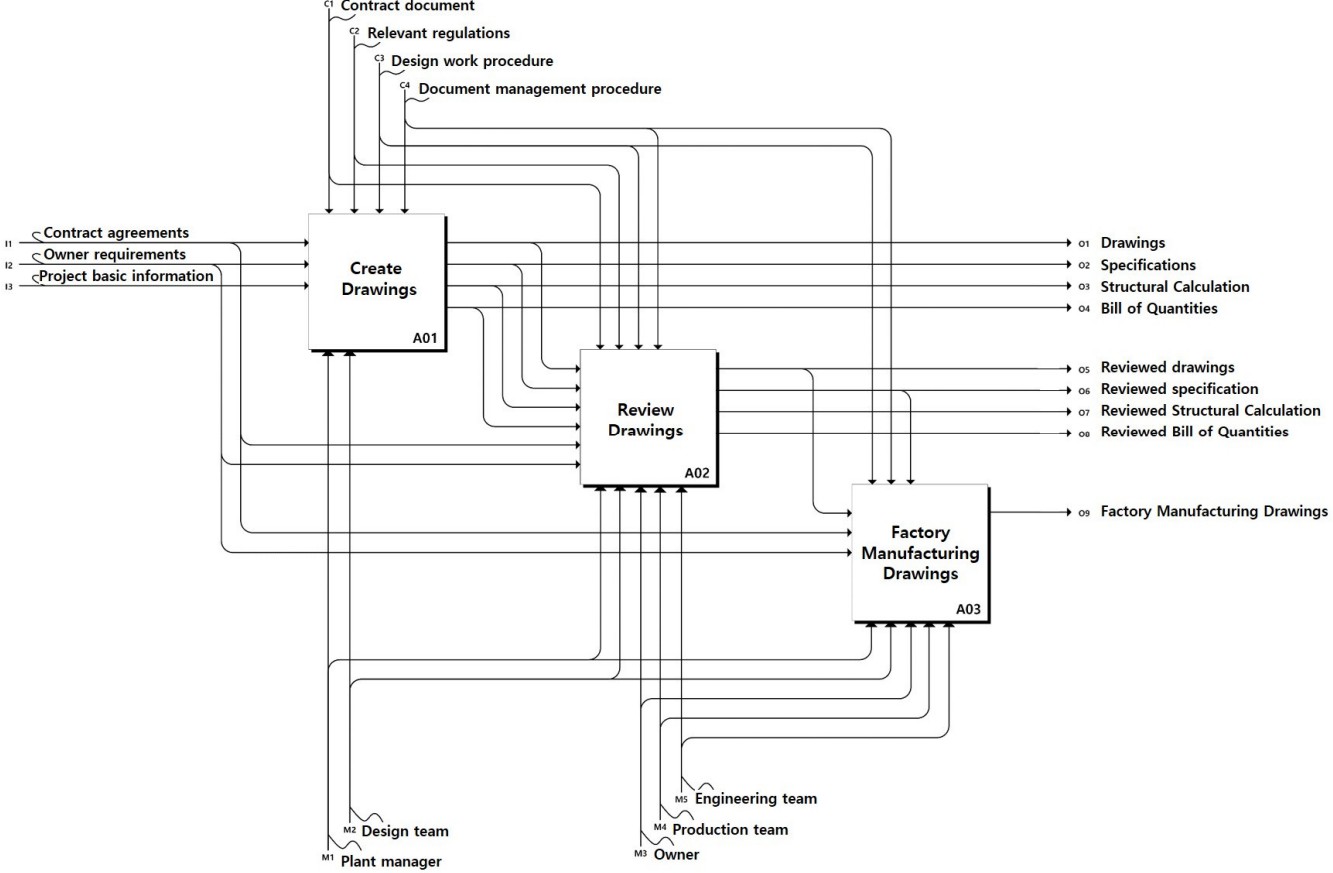

**Figure 3.** IDEF0 modeling results for quality management in design.

### 3.2.2. Material Information Standardization

Figure 4 shows the standardization of quality information on materials during module manufacturing. Material-related quality management tasks can be expressed in the following steps: (B01) establishment of material supply plan, (B02) approval of material suppliers, (B03) acceptance inspection of materials, and (B04) inspection of imported materials. The first three are conducted in the pre-preparation stage before module production, and B04 is conducted during the module production stage. Generally, the material supply plan is established according to the basic project information, such as the requirements of the owner and the project outline. The type of material for manufacturing the module depends on the design drawings and specifications. Even if the material type has been determined, the material import schedule will depend on the overall module manufacturing schedule. Therefore, establishing a supply plan for equipment and material follows the initial stage of design and manufacturing. In it, basic information on imported materials, such as item, specification, unit, quantity, and import schedule, is created. Depending on the equipment and material supply plan, major equipment and materials that require supplier approval are selected. Major equipment and material are those that directly affect the performance of modular buildings, such as concrete and structural steel. Supplier approval involves reviewing and determining whether the supplier can supply the materials required in the design documents. Therefore, different qualification certificates and information on material performance certification that can prove stable delivery on the part of the material supplier are primarily managed in this stage. After supplier approval, the materials go through a pre-acceptance inspection before entering the factory. Acceptance inspection is a repeat inspection before entering the factory to check the performance of key materials, where the final performance suitability of the purchased material is judged. It mainly involves checking whether the conditions of incoming materials and suppliers match the

supporting documents. If there is no major problem in the material performance, the material that has passed the acceptance inspection is brought into the factory. The imported materials go through a material inspection process before being utilized in module manufacturing. Here, a checklist containing the main inspection items for each material is used, and whether the material is damaged or matches the order information is determined. Further, information such as incoming invoice information and material inspection results is generated.

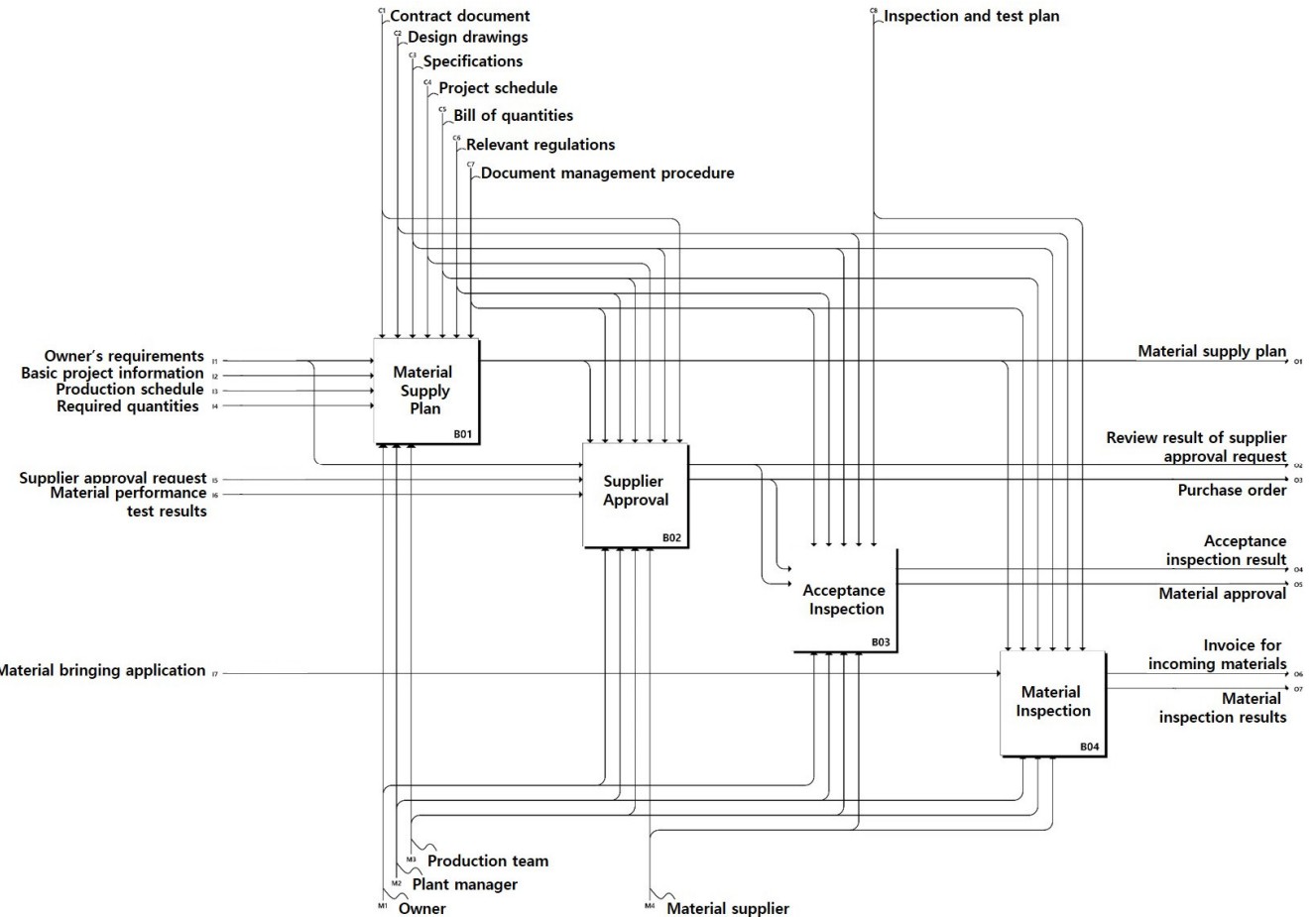

**Figure 4.** IDEF0 modeling results for quality management in material management.

### 3.2.3. Standardization of Construction Information

Figure 5 shows the results of standardization of construction quality information during module manufacturing. Construction-related quality management tasks can be represented as follows: (C01) production plan establishment, (C02) inspection and test plan establishment, and (C03) construction inspection and test steps. C01 and C02 are conducted in the pre-preparation stage before factory production. C03 is performed at the module manufacturing stage. Generally, the module manufacturing plan depends on basic project information on the owner's requirements, construction outline, etc. The module production plan can be established after the basic design document is prepared. A module manufacturing plan is established based on the contents of the prepared basic design documents, considering the status of the workers and the equipment that can be used in the factory. In the production plan establishment stage, what production line and equipment to use in the factory manufacturing process, how many workers to employ, how to plan the movement line, how to set the production schedule for each stage of construction, etc., are decided. Therefore, information about the production schedule and construction and manufacturing plans is generated. After the module manufacturing planning stage,

an inspection and test plan is established according to the established manufacturing schedule. Inspection and testing are conducted for the main processes that affect the performance and quality of the module. In the inspection and test plan establishment stage, the process that requires inspection and testing is selected, and the inspection and test method, schedule, and main items of the selected process, are determined. Based on the relevant regulations and project specifications, some tests and inspections are planned to be conducted by accredited laboratories, while the others are planned to be performed at the factory. This process generates information on construction inspection checklists and quality test plans for the critical processes. At the module manufacturing stage, construction inspection and quality testing are conducted in accordance with the established plan. Construction inspections and tests are based on the main inspection items, methods, and procedures determined during the inspection and test plan establishment stage. In this step, information is generated on the quality performance of each module, such as the construction inspection and test report or results of quality tests reported by accredited laboratories.

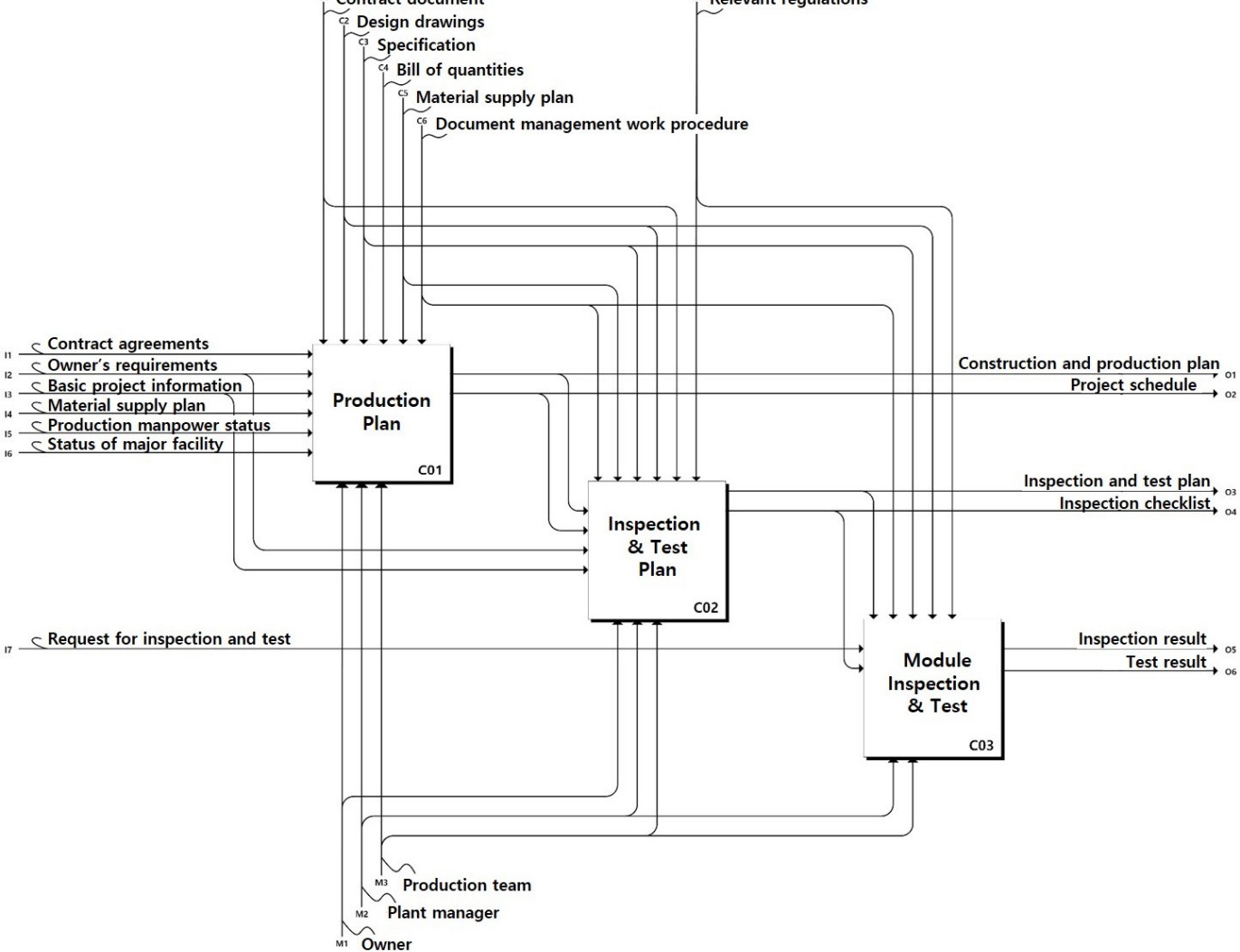

**Figure 5.** IDEF0 modeling results for quality management in construction.

### 3.3. Defining the Main Functions of the System

System function definition aims to facilitate system development by clarifying the purpose of the function, standard operation method, and scope of the development work [32]. By defining key functions before system development, unnecessary functions in the devel-

opment process can be eliminated, and the usability of the system can be improved [33]. The system function is defined after defining the main function, which comes after the basic function is defined. The menu of the development system was configured according to the function definition result. The system configuration menu is shown in Table 2.

**Table 2.** Modular Factory Quality Information Management System Menu.

| Main Menu | Sub Menu | Purpose of Development |
|---|---|---|
| 1. Basic Project Information | 1-1. Basic information and General Items | Project basic information management Manufacturing module information management per project |
| | 1-2. Project module information | |
| | 1-3. Module-type management | |
| | 1-4. Production process management by module type | |
| 2. Design management | 2-1. Design document review | Design document management Design document establishment/revision management |
| | 2-2. Factory manufacturing drawing review | |
| 3. Material management | 3-1. Equipment and material supply plan | Material-related document management Material acceptance and inspection management |
| | 3-2. Inspection and test report | |
| | 3-3. Supplier approval | |
| | 3-4. Incoming material inspection | |
| | 3-5. Management of nonconforming material | |
| 4. Manufacturing management | 4-1. Manufacturing inspection management | Module manufacturing inspection and test management |
| | 4-2. Nonconforming manufacturing management | |
| | 4-3. Module inspection progress | |
| | 4-4. Test management | |
| 5. Document management | - | Management of documents related to quality control |
| 6. System management | 6-1. User management | System database management |
| | 6-2. Test equipment management | |
| | 6-3. Project management | |
| | 6-4. Checklist management | |
| | 6-5. Admin menu | |

The system basic function aims to manage basic information, such as project basic information. The general information of the modular project can be managed through the project basic information management function. Specifically, information such as project outline, quality policy, quality management procedure, and module information can be managed. Module information includes where each module is installed in a building (i.e., building, floor, and room number) and how it is manufactured. The user can more intensively manage the module information through the independent module information management function. On the other hand, general project information, such as construction outline, quality policy, and quality management procedures are all composed at a basic level. Therefore, it would be more efficient to manage the general project information as a single function.

The main function of the system is to manage the quality information related to design, materials, and construction in module factory production according to the information standardization result. The design management function aims to manage the quality information related to the design of the modular project. Design-related quality information includes the revision history of design documents, review opinions, and special matters in the review and revision stage. The design management function allows for efficient tracking and management of design document information. It involves separating basic

design documents and factory manufacturing drawings according to the type of design documents, thus improving the identification of design quality information. System users can systematically manage design document information through the design management function. Furthermore, it is possible to prevent problems related to the quality of design documents, such as discrepancies between design documents, because sharing work contents with other fields is simplified.

The material management function aims to manage material information used in module manufacturing and quality information related to material supply. It separates detailed functions for material supply planning, supplier approval, and material inspection. To track and manage material-related quality information easily, the revision history information of each material supply plan is included. The source approval function handles information about the qualifications of major material suppliers. Therefore, information such as the quantity of each material brought in, specifications, suppliers, and various supporting documents related to material performance, are included. The material inspection function manages quality inspection information for materials brought into the modular factory. It uses the existing checklist in the system database. Accordingly, a function for editing the material checklist should be separately prepared to increase the system usability. In addition, nonconformities that occur during material inspection should be managed through a separate and independent function to effectively utilize material defect information in future projects.

Manufacturing management targets module manufacturing inspection and test information management. The module manufacturing inspection function manages inspection information for each module manufacturing process. Further, manufacturing inspection utilizes the checklist included in the system database in advance, and the user can edit the checklist, similar to in the material inspection function. Meanwhile, when a large number of modules are manufactured, determining the inspection progress of each module may be difficult. Accordingly, a function should be prepared to provide information on the module inspection progress of the entire modular project, see Table 2—Modular Quality Information Management System Configuration Menu.

## 4. Development of Quality Information Management System for Modular Construction Factory

### 4.1. System Overview

The developed system was planned to be web-based to increase the user's system accessibility. It consists of a client that needs and a server that provides information. A MySQL server was used for the modular factory quality information management system. MySQL is advantageous because it can be used on various operating systems, such as Unix, Linux, and Windows. The MySQL server was used to prepare for future server migration and system expansion. For system deployment and server operation, Docker—an opensource virtualization platform based on containers—was used. A container packages and isolates software, such as libraries, system tools, code, and executable files, for system operation. Using Docker can reduce the hardware resources required for server operation, thereby reducing the server operation burden on the host.

### 4.2. Main System Features

4.2.1. Module Information and Production Type Management

To reduce the defect rate in the module manufacturing process, it is important to clearly identify the target defect and its cause. In large projects where many identical modules are built, it is difficult to identify and track individual modules. The developed system manages module information by dividing it into information about each module and information about the module type. Module information was divided based on module ID, building, floor, room number, and use to prevent duplicate management of modules. Generally, modules manufactured in a modular project are constructed differently, depending on the structural systems, material type, and construction methods applied to

each module. Therefore, changes in module types lead to changes in key quality check items for the module manufacturing process. Therefore, the modular factory quality information management system was developed to classify and manage module types to effectively manage module production inspection information. Therefore, 86 modular building projects undertaken in Korea from 2003 to 2020 were analyzed to identify and predefine the different types of modules. First, the module types were classified by examining the commonly applied construction methods regardless of the module type and analyzing the construction methods involving different construction processes. Therefore, depending on the type of slab construction method (wet or dry floor), type of fireproof construction (fireproof painting, fireproof spraying, or encasement fireproofing), type of toilet construction (wet or dry toilet), and heating construction (wet or dry heating), 24 module types were predefined. In addition, because the predefined module types may not be comprehensive, the developed quality information management system also includes a function to create and manage module-type information. The user can set the types of slab construction, fireproof construction, toilet construction, and heating construction for each module using the module type management function. Information on the module type is registered in the system according to the set construction method, and the module manufacturing process and main inspection items of each process are determined for each registered module type. In addition, by designating module type information to each previously created module, inspection items and contents for each module can be used in the production inspection step. Figure 6 shows the process of assigning the created module type to the created module information.

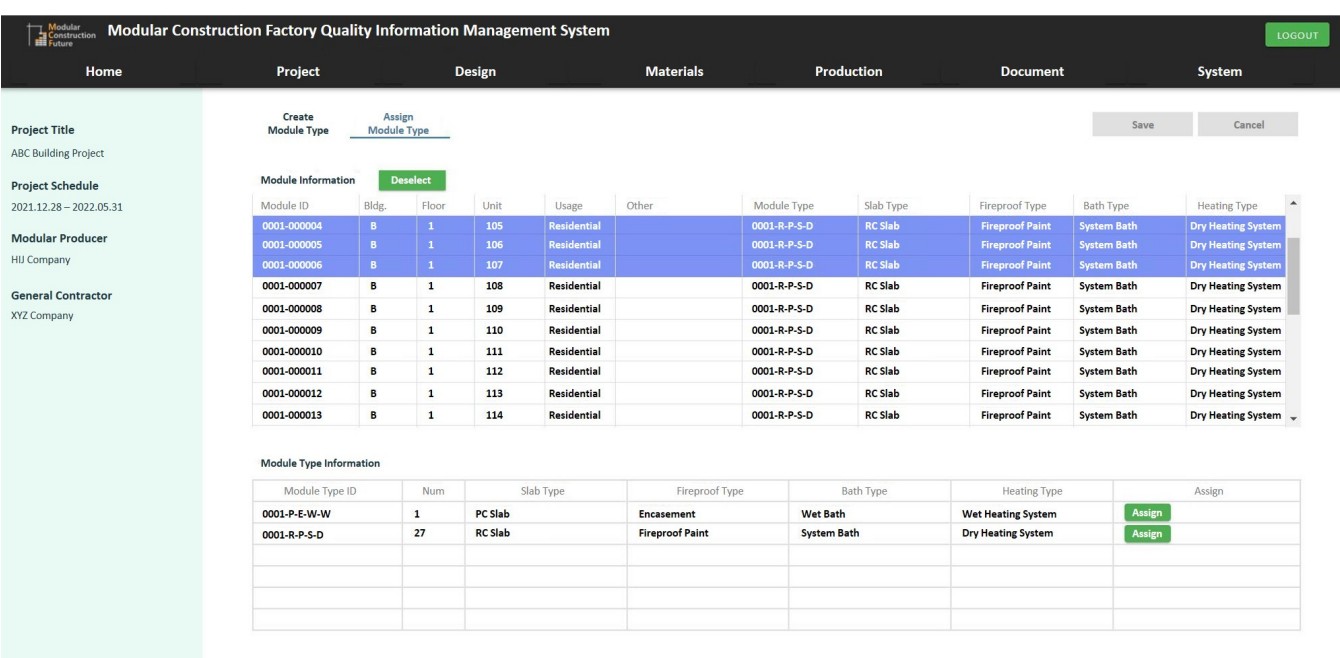

**Figure 6.** Module type information designation function.

### 4.2.2. Material Management

The quality information management system can track and manage material quality information from the initial planning stage to the final construction stage. Specifically, the material management function manages material quality information in the factory pre-production stage through equipment and material supply plan management, inspection, and test plan management, supplier approval, and incoming material inspection functions. The equipment and material supply plan management and inspection and test plan management functions were developed to manage material-related information in the planning stage. Users can manage the establishment/revision history of each plan

through equipment and material supply plan management and inspection and test plan management functions and share the review with other stakeholders. The supply approval function was developed to manage in detail the main equipment and material information selected in the planning stage. The supplier approval function manages basic material information such as material items and specifications and information on documents related to supplier approval. Specifically, the source approval function works by inputting information about the main material and uploading/downloading the source approval application document. The incoming material inspection function manages the factory inspection information of key materials, and it is presented in Figure 7. The incoming inspection of each material is performed using the checklist for each material stored in the system database. The user can add or edit the contents of the material inspection checklist through the system management function of the quality information management system. Nonconformities identified during the inspection of incoming materials are managed separately in the management of nonconforming equipment function. In it, information such as nonconforming material nonconformity, nonconforming action details, and quantity of nonconforming material, is managed.

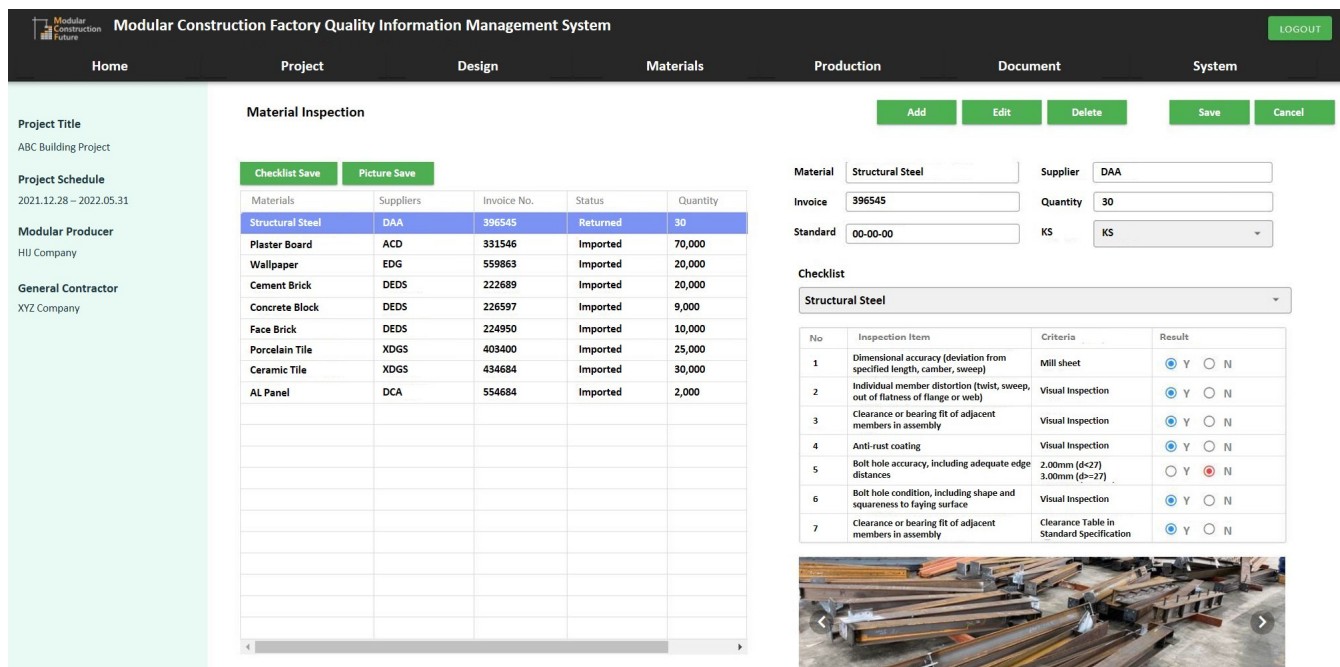

**Figure 7.** Material inspection information management function.

### 4.2.3. Module Manufacturing Management

The module production management function can manage test information and production inspection information generated during the production of each module. Module test management was developed to manage various test information to ensure module performance, such as fire resistance, waterproofness, and airtightness. The user can manage information such as test types, standards, and scores through the module test management function. Module manufacturing inspection management was developed to manage manufacturing inspection information for each process according to each module type. As mentioned above, the developed system includes data on the manufacturing process and a step-by-step inspection checklist according to 24 predefined module types. Similar to material inspection management, the checklist for manufacturing inspection can be added and edited by the user. Production inspection is conducted on the entire module input in the module information management function. The module manufacturing inspection management function is shown in Figure 8. The information of all modules included in the project and information of the module selected for inputting the current inspection

result are displayed at the top of the production inspection function screen. The process status of the module is displayed on the left side of the screen. The diagram schematizes the production process and inspection progress of the module. When all inspection items in the process are completed, the inspection items in the next process are displayed. The manufacturing inspection status indicator appears in the upper-right corner of the diagram, and the process under inspection is displayed in black; when all inspection items are completed without nonconformity, in green; and when one or more nonconformities occur, in red. The user can check the production and inspection status of individual modules through the module manufacturing inspection management function.

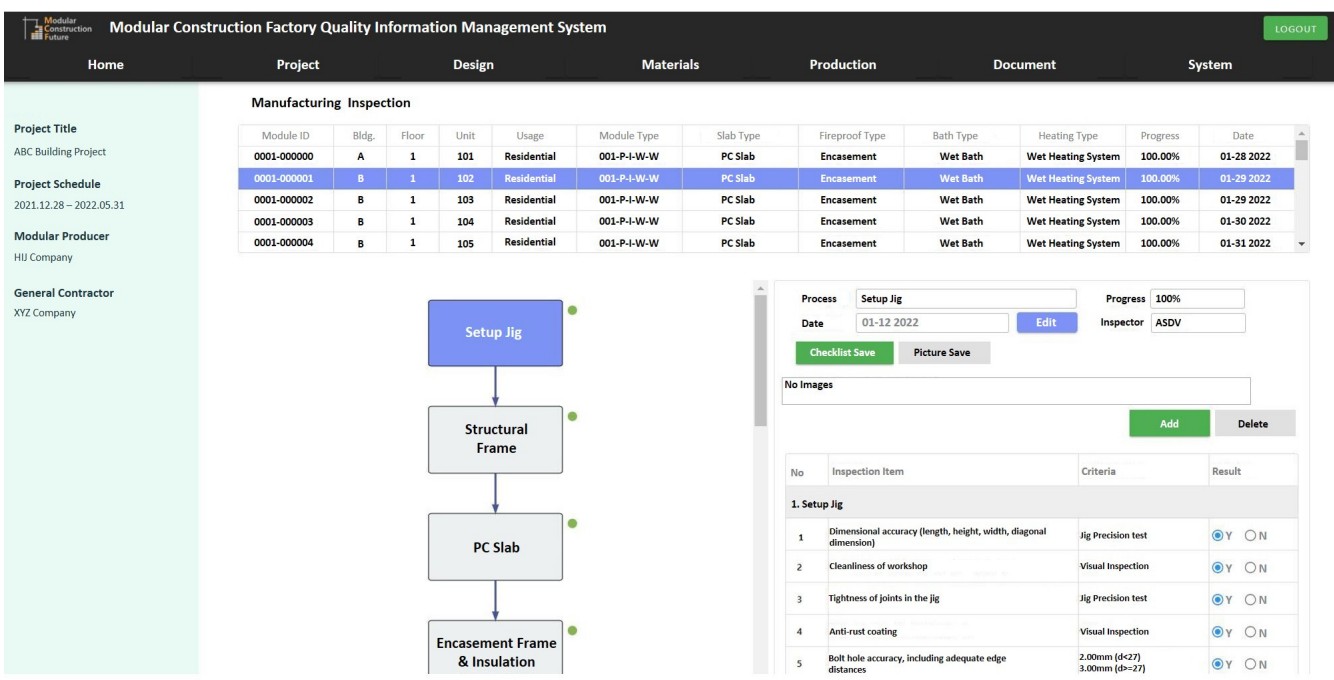

**Figure 8.** Module production inspection management.

Nonconformities that occur during manufacturing inspection are separately managed through the management of the nonconformity manufacturing function. The quality information management system recognizes a process as nonconforming when one or more of the checklist items used for manufacturing inspection indicate nonconformity. Each item in the checklist has an error tolerance for determining nonconformity. These tolerances are determined by the type of work, relevant regulations, and specifications. The developed system is designed to accept error tolerance inputs from users. When users upload inspection checklists based on the inspection and test plan, they are asked to determine the error tolerance for each item in the checklist based on the type of work, relevant regulations, and specifications. Then, the users refer to the error tolerances in the system when conducting inspections and tests during manufacturing. The information of the module in which the nonconformity occurred is displayed at the top of the screen of the nonconformity production management function. In it, detailed information about the process in which nonconformities occurred is displayed on the left side of the screen, and the reason for nonconformity and measures to be taken are displayed on the right side. The nonconforming product management function is shown in Figure 9.

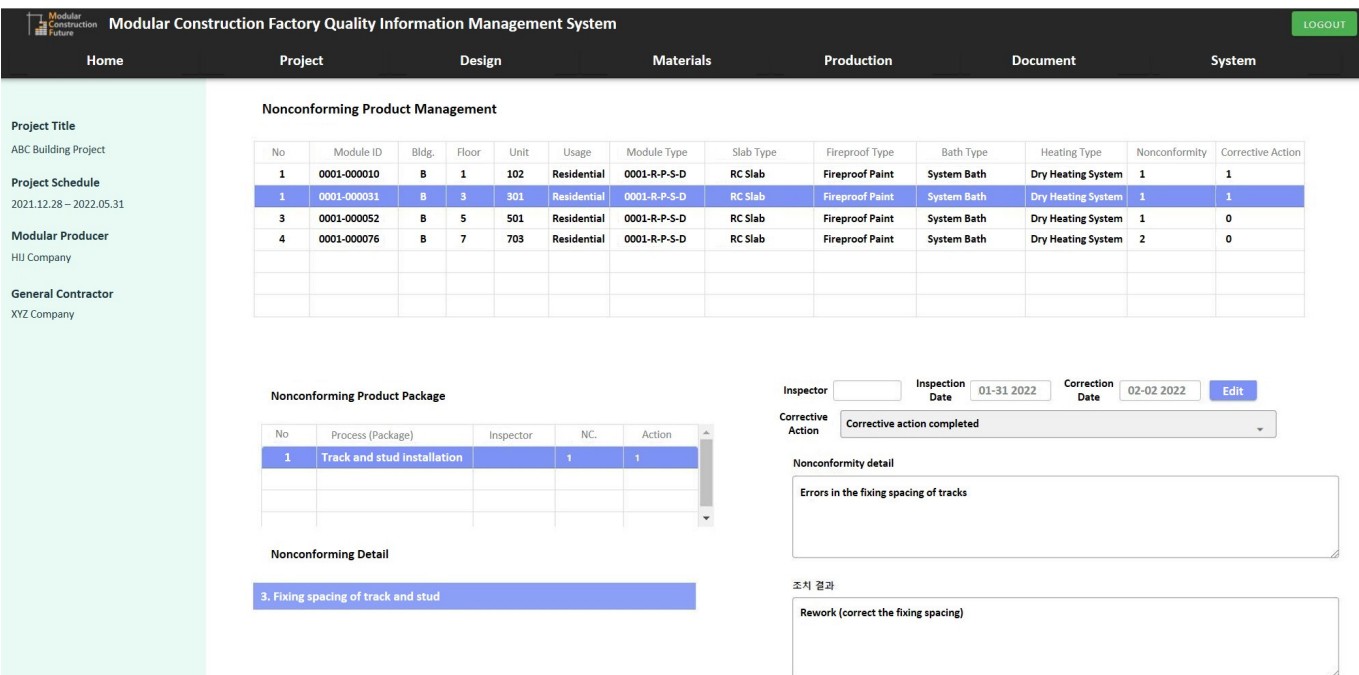

**Figure 9.** Management of nonconforming module information.

Although the module manufacturing inspection management function provides information on the manufacturing inspection status of each module, it is difficult to determine the overall module manufacturing inspection status in a modular project that manufactures multiple modules. This difficulty is exacerbated when modules of multiple projects are produced in one factory. To resolve this, a function to visualize the overall module production inspection process of the project was developed for the quality information management system. The module inspection status management function is shown in Figure 10. It displays the overall module production inspection status of the modular project, inspection history of individual modules, and Takt Schedule Chart. In the module inspection status management function, the inspection status information of the entire project module is displayed at the top of the screen, and the inspection history information of the selected module is displayed at the bottom. The detailed production process for each module depends on the previously set module type. To determine the inspection status of all modules irrespective of module type, this study investigated the common processes in all module types. Accordingly, the module manufacturing process could be divided into eight main processes: structure assembly, slab construction, fireproof construction, wall construction, toilet construction, ceiling construction, floor heating construction, and final finishing construction. Therefore, the bar chart displaying the module inspection status is composed of eight steps. The production inspection status management function intuitively describes the overall production and inspection status of the project module to the user. The production inspection status management function is also useful in terms of schedule management; the user can check the module production status through the information provided and adjust the production schedule accordingly. The module manufacturing inspection status management function facilitates the tracking and management of the quality information of each module by displaying the detailed inspection history of each module along with the status diagram.

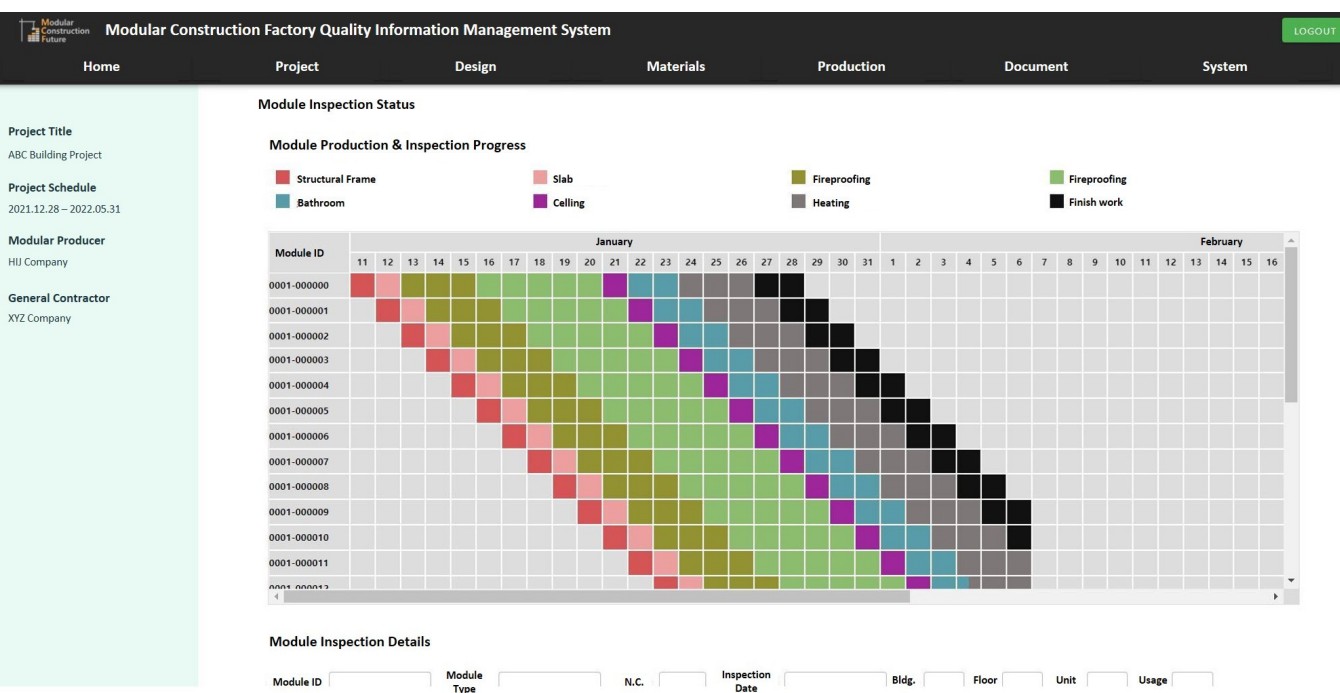

**Figure 10.** Information management in the production inspection status.

## 5. System Evaluation

System evaluation is critical for determining the practicability of a system [34]. It provides feedback on functional improvement and supplementation. In this study, system evaluation was conducted to verify the usefulness of the development system and to confirm its practicability. It can be divided into system performance evaluation in terms of implementation of system functions and system usability evaluation and in terms of implemented system functions according to the purpose of evaluation [35]. The usability of the developed system can be measured through system performance and system usability evaluations. Further, it is necessary to verify whether the system developed is practically helpful for quality control in module production. Therefore, this study conducted a system evaluation targeting modular experts.

### 5.1. System Performance Evaluation

The system performance evaluation aims to evaluate whether the system can normally provide the functions required by the user under the planned operating environment [36]. In this study, validation and verification test (V&V test) by accredited testing and certification laboratory was implemented for objectively evaluating system performance. The certified performance test was conducted for all major functions except for the system management function. The accredited performance test was conducted in accordance with ISO/IEC 25023:2016, an international software quality standard. It was developed to quantitatively evaluate the quality of a system using the ISO/IEC 25023:2016 standard [37], and it provides a test method and performance criteria suitable for the function to be evaluated [38]. The performance test of the modular factory manufacturing quality information management system was conducted by checking whether the main functions of the system operate normally according to the defined operation method. Table 3 shows the official performance test results of the developed quality information management system. The test system execution result matched the expected result, and the system function operates normally and as planned.

**Table 3.** Developed system performance test results.

| No | Detail Function | Expected Result | Implementation Result | |
|---|---|---|---|---|
| | | | Input/Save | View |
| 1 | Design management | - Stable upload/download of basic design drawings and factory manufacturing drawing documents.<br>- The history of establishment and revision of basic design drawings and factory manufacturing drawings is normally reflected in the system. | Expected result Satisfaction | Expected result Satisfaction |
| 2 | Material management | - Product name, specification, and verifying document information are normally reflected in the system<br>- Stable linkage with nonconforming material management function.<br>- Stable linkage with nonconforming material management function. | Expected result Satisfaction | Expected result Satisfaction |
| 3 | Module information management | - A unique module number (module ID) is generated, and basic information for each module (building/floor/room number, usage) is not duplicated.<br>- The entered module information is normally registered in the system. | Expected result Satisfaction | Expected result Satisfaction |
| 4 | Module type management | - All types of process diagrams are implemented normally.<br>- All types of inspection checklists are normally viewed.<br>- Module information and manufacturing type information designation works normally. | Expected result Satisfaction | Expected result Satisfaction |
| 5 | Production inspection management | - All module information registered in the system and the inspection process of each module are displayed on the screen.<br>- The status indicator of the inspection process works normally.<br>- Stable linkage with nonconforming product management functions. | Expected result Satisfaction | Expected result Satisfaction |
| 6 | Module production inspection status | - The module inspection status is normally displayed according to the entered manufacturing inspection result.<br>- The detailed history of the selected module is displayed normally. | Expected result Satisfaction | Expected result Satisfaction |

*5.2. System Usability Evaluation*

System usability evaluation aims to determine the usability of the system from the user's point of view [39]. It can be used to judge the usefulness and reliability of various objects, such as general devices, web search engines and maps, application software, and systems [40]. System usability evaluation is divided into analytical and empirical evaluation according to the evaluation subject. The former is also called predictive evaluation because it is conducted by experts and predicts problems that may occur when using the system. It includes list test, rehearsal test, model analysis, and simulation. An empirical evaluation is conducted by real system users. An empirical evaluation is mainly conducted in the form of post-evaluation for a certain period after the system is launched.

In this study, heuristic evaluation, one of analytical evaluation methods, was conducted to check the usability of the developed quality information management system. It is mainly performed to evaluate system usability in terms of user interface and design of the developed system [41]. As a representative method, it is used for system usability evaluation due to cost and time advantages and efficiency [42]. It is conducted by three to five experts writing an evaluation sheet on the Nielsen [43]'s 10 principles of the heuristic.

The questionnaire was developed based on a Likert 5-point scale (1 point: disagree, 5 points: agree). The heuristic evaluation items were reconstructed according to the characteristics of the modular factory quality information management system and existing heuristic evaluation studies. Accordingly, a survey with 21 evaluation questions to evaluate seven items of system status visibility, match the system and real world, evaluate user control and freedom, consistency and standard, error prevention, recognition rather than recall, and aesthetic and minimalist design was developed. A questionnaire response sheet with questions was also developed. Table 4 shows the contents of the developed questionnaire response sheet. The heuristic evaluation of the developed quality information management system was conducted by two experts in the software field and three experts in modular architecture. For the evaluation, the characteristics of the development system and the main purpose of each evaluation item were first explained to the survey respondents. To help them understand the system, the system user manual and main function demonstration video were provided. For the analysis of the evaluation results, they were explained for negative responses or non-evaluable items. All items were found to be generally excellent (mean = 4.43, SD = 0.610), particularly the match between the system and real-world items (mean = 4.68, SD = 0.471). However, the user control–freedom item received a relatively low evaluation (mean = 4.06, SD = 0.524).

**Table 4.** Contents of the questionnaire for heuristic evaluation.

| Heuristic Evaluation Items | Evaluation Questions |
|---|---|
| Visibility of system status | Can the user know at a glance the current state of the system and what operations are currently available just by looking at the screen? |
| | Are the current states clearly marked, such as icons, images, or hypertext? |
| | Is it clearly indicated which location the user has currently selected? |
| Match between system and the real world | Are the terms used in the input window frequently used by users? |
| | Does the form on the document that users work on match the form on the computer? |
| | Are highly related items appearing on the same screen? |
| User control and freedom | Is there an undo function for each action? |
| | Can the user go back to the previous menu item and change a selection already made? |
| | Is the button to the home page on each screen prominently displayed? |
| Consistency and standards | Are the names of the same menu items presented consistently within a system? |
| | Is the length of the term appropriate? |
| | Do all pages have titles and headers that describe their content? |
| Error prevention | Is it possible to simultaneously input letters and numbers in one field in the input window? |
| | Are the buttons that are not currently applicable are dimmed or not shown at all? |
| | Does the system warn the user of the consequences before executing a function that could have serious consequences? |
| Recognition rather than recall | Are the infrequently used but essential tasks easy to remember in order? |
| | Do the menu items provide multiple steps for the user to remember with ease? |
| | Are the names of the buttons clear and easy to understand? |
| Aesthetic and minimalist design | Are the buttons refraining from overly detailed expressions in the button design? |
| | Is it refraining from using too many colors? |
| | Are colors used to make it easier to distinguish between text and background color? |

## 5.3. System Utility Evaluation

System usability evaluation aims to determine whether the efficiency of the production quality management work in the modular construction factory is improved by the

developed system [44]. To derive meaningful evaluation results, practical application is necessary. However, this is difficult because the application is in the precommercial stage. Therefore, this study evaluated the usefulness of the system by interviewing modular construction experts. Five modular construction experts and five research and development experts on modular construction participated in the system utility evaluation. All modular construction experts have more than seven years of experience in module manufacturing (mean = 10.6, SD = 2.65). Moreover, all research and development experts have more than four years of experience (mean = 6.4, SD = 1.62) in modular construction research. Prior to evaluation, each expert received a preliminary explanation on the system function configuration and key functions (i.e., project module information management, module type management, production process management by type, supply source approval, material incoming inspection, nonconforming equipment management, manufacturing inspection management, nonconformity production management, and module inspection status), and watched a demonstration video. After performing the system function according to the directed request, the usefulness of the main function of the system was evaluated using a 7-point Likert scale (1 point: not very useful, 7 points: very useful). In addition, opinions on system improvement were collected using open-ended questions.

The evaluation revealed that the system was generally useful for most functions (mean = 6.01, SD = 0.641), and production inspection management was evaluated as the most useful (mean: 6.4, SD = 0.490). It received a high evaluation in terms of practical utility because manufacturing inspection is repetitive and frequently occurs during quality control activities in the module factory. The module inspection status was evaluated to be the second most useful (mean = 6.3, SD = 0.458), presumably because it helps to intuitively determine the overall project progress status in connection with process management. In addition, two additional functions were developed based on expert opinions on system improvements. First, to improve the linkage with existing quality management work, a function was added to automatically output the results of incoming material and manufacturing inspection in the form of an existing quality result report. This is expected to increase the utility of the developed system in the quality control of the current module factory. In addition, the improvements to the developed system can allow it to be used flexibly according to project conditions by allowing users to add and modify module types and checklists, depending on project characteristics in module-type management and varied-checklist management.

## 6. Conclusions

The importance of quality information in the construction industry is increasing due to the increase in the scale and complexity of construction projects. Although the usefulness of quality information is greater for modular construction because it has the characteristics of a manufacturing industry, studies on quality information management during factory production have rarely been conducted in contrast to the case of site-oriented construction. In this study, we developed a quality information management system that can effectively manage and utilize quality information generated during manufacturing in a modular factory. The module manufacturing process was analyzed to determine the characteristics of modular construction in the system function. To manage the quality information generated in each quality management process in an integrated system, information standardization was conducted using IDEF0. From the information standardization, the system function was defined and system development conducted. The developed quality information management system efficiently manages the quality information generated during manufacturing in the factory. Representative functions of the developed modular quality information management system include module information and production-type management, material management, and module production management. Using the module information and production type management function, the utilization of the module production quality information can be improved. Using the material management function, it is possible to track the quality information of the materials used to manufacture the

module. Using the module production management function, the production inspection information for each module and the production status of all modules can be interpreted.

The developed quality information management system can improve the inefficiency in the quality management process of module manufacturers by providing an integrated and systematic method to manage quality information generated during manufacturing. The developed system can prevent the generation of redundant quality information and improve identification and traceability by recording quality information management history. Furthermore, the information provided by the developed system, such as production inspection status information, can be utilized in tasks other than quality management (e.g., production schedule management). Therefore, if the development system is applied, the overall efficiency of quality management tasks in the module manufacturing process can be increased. In this study, performance and usability evaluation were performed to confirm the practicability of the module factory manufacturing quality information management system. The evaluations were conducted through a V&V test and heuristic evaluation, respectively. The performance evaluation indicated that all functions of the developed quality information management system operated normally as predefined. The evaluation of the usability and usefulness of the system revealed that the developed system had excellent usability, and its practical usefulness was also judged to be high. However, accurately reflecting the positions of all practitioners in the evaluation of system usability by interviewing experts had limitations. Accordingly, future research should evaluate the usefulness of the development system based on the results of practical application over a duration. In addition, it is necessary to prepare a plan to maximize the advantages of the system from the practitioner's point of view and to compensate for the disadvantages of the system. The system performance can be improved by incorporating the suggestions of practitioners. The improved system will have a positive effect on the quality of the manufactured module as well as on the quality control.

**Author Contributions:** Conceptualization, J.S. and B.C.; methodology, J.S. and B.C.; software, J.S.; validation, J.S. and B.C.; formal analysis, J.S. and B.C.; investigation, J.S.; data curation, J.S.; writing—original draft preparation, J.S.; writing—review and editing, B.C.; visualization, J.S.; supervision, B.C.; project administration, B.C.; funding acquisition, B.C. All authors have read and agreed to the published version of the manuscript.

**Funding:** This research was supported by the Housing Environment Research Program funded by the Ministry of Land, Infrastructure, and Transport of the Korean Government [21RERP–B082884-08, 2021].

**Institutional Review Board Statement:** Not applicable.

**Informed Consent Statement:** Not applicable.

**Data Availability Statement:** The data presented in this study are available on request from the corresponding author.

**Conflicts of Interest:** The authors declare no conflict of interest.

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
