# Peer review of "Design and Implementation of Quality Information Management System for Modular Construction Factory"

_buildings, doi:10.3390/buildings12050654_

Round 1
Reviewer 1 Report
The manuscript presents a really well-designed research with sound methodology and execution. The topic is of great interest to the community and can certainly have practical implications. My only remarks are related to clarification of certain aspects of the model/system, which can be found in the comments in the attached document.

Author Response
We appreciate the time and efforts to review the paper and provide constructive feedback. We have carefully considered your comments and revised the manuscript based on every comment.

Reviewer 2 Report
- Add a full stop at the end of the sentence at line 23.
- Remove the full stop after the sentence at line 48.
- The authors may consider changing the sentence at lines 60-62 from past tense to present tense. Consistency is the goal here.
- The authors have declared that research on quality information management is insufficient and that there is a need to develop a quality information management system (QIMS) for manufacturing companies operating in the domain of modular construction. Can the authors categorically state that the QIMS developed in this manuscript is one of a kind and no frameworks for a QIMS have ever been developed before for any manufacturing industry? If not, what is the novelty in this manuscript? The authors may consider explaining this in the introduction section.
- The authors have argued in the abstract that “… the existing inefficient quality management process of module manufacturers …”. The authors may consider adding some evidence in the text to support this assertion on inefficiency of the existing quality management processes.
- Background information on companies interviewed, experience profile of the interviewees, and number of interviews, are not provided.
- How can one be sure that any QIMS is not being used by other modular construction companies? How can we be sure that the information that was fed in for the development for the QIMS is reflecting input from a reliable sample among the population of the modular construction companies? The sample seems to be too small. The authors may consider justifying the sample size adopted in this work.
- Replace IDEF with IDEF0 at line 203.
- Replace “In” with “It” at the start of the sentence at line 271.
- Font size of text in figures 3, 4, and 5 may be resized to enhance legibility. Resolution of the remaining figures may also be enhanced.
- The word “and” in the statement “It consists of a client that needs and a server that provides information.” at line 374 may be redundant.
- The phrase “please add:” may be removed at line 657.
Author Response

(The authors gave the same response as above.)
